
# Analysis of Conformational Exchange Processes using Methyl-TROSY-Based Hahn Echo Measurements of Quadruple-Quantum Relaxation

Christopher A. Waudby[1] and John Christodoulou[1,2]

[1]Department of Structural and Molecular Biology, University College London, London WC1E 6BT, UK
[2]Francis Crick Institute, London NW1 1AT, UK

*Correspondence to*: Christopher A. Waudby (c.waudby@ucl.ac.uk)

**Abstract.** Transverse nuclear spin relaxation is a sensitive probe of chemical exchange on timescales on the order of microseconds to milliseconds. Here we present an experiment for the simultaneous measurement of the relaxation rates of two quadruple-quantum transitions in $^{13}CH_3$-labelled methyl groups. These coherences are protected against relaxation by intra-methyl dipolar interactions, and so have unexpectedly long lifetimes within perdeuterated biomacromolecules. However, these coherences also have an order of magnitude higher sensitivity to chemical exchange broadening than lower order coherences, and therefore provide ideal probes of dynamic processes. We show that analysis of the static magnetic field dependence of zero-, double- and quadruple-quantum Hahn echo relaxation rates provides a robust indication of chemical exchange, and can determine the signed relative magnitudes of proton and carbon chemical shift differences between ground and excited states. We also demonstrate that this analysis can be combined with established CPMG relaxation dispersion measurements, providing improved precision in parameter estimates, particularly in the determination of $^1H$ chemical shift differences.

## 1 Introduction

Solution NMR spectroscopy is a powerful tool for the characterization of macromolecular dynamics over a range of timescales relevant to biological function (Sekhar and Kay, 2019): from backbone and sidechain disorder on ps–ns timescales, characterized by $S^2$ order parameters (Frederick et al., 2007; Stetz et al., 2019; Sun et al., 2011); rotational diffusion and domain motions, characterized by rotational correlation times, $\tau_c$ (Ryabov et al., 2009; Waudby et al., 2021); through to real-time measurements of kinetics on timescales of seconds and beyond, following rapid mixing, temperature or pressure jumps (Charlier et al., 2018; Franco et al., 2017; Waudby et al., 2018). NMR is also particularly well suited to the analysis of reversible chemical exchange on timescales of the order of microseconds to milliseconds, via lineshape analysis across a titration series (Stadmiller et al., 2020; Waudby et al., 2016, 2020) or using sophisticated pulse sequences such as ZZ-exchange spectroscopy, chemical exchange saturation transfer (CEST), and Carr-Purcell-Meiboom-Gill (CPMG) and $R_{1\rho}$ relaxation dispersion (Alderson et al., 2020; Boswell and Latham, 2019).



An NMR-active spin in chemical exchange between two conformations experiences an additional contribution, $R_{ex}$, to its observed transverse relaxation rate:

$$R_2 = R_{2,0} + R_{ex} \tag{1}$$

where $R_{2,0}$ is the relaxation rate in the absence of exchange. For exchange between two conformations, labelled A and B, the magnitude of $R_{ex}$ depends on the populations, $p_A$ and $p_B$, the exchange rate, $k_{ex} = k_{AB} + k_{BA}$, and the difference in

frequency of the observed coherence between states A and B, $\Delta\omega$. For a single quantum coherence of nucleus X this frequency difference, $\Delta\omega_X = \gamma_X B_0 \Delta\delta_X$, depends on the gyromagnetic ratio of the nucleus, $\gamma$, the chemical shift difference, $\Delta\delta$, and the magnetic field strength, $B_0$; while for multiple quantum coherences, frequency differences reflect linear combinations of the individual coherences. In the fast exchange limit, the focus of much of this paper, the exchange contribution to relaxation for a single quantum coherence of nucleus X is:


$$R_{ex} = \frac{p_A p_B \Delta\omega_X^2}{k_{ex}} = \xi_X^2 B_0^2 \tag{2}$$

where:

$$\xi_X = \sqrt{\frac{p_A p_B}{k_{ex}}} \gamma_X \Delta\delta_X \tag{3}$$

represents the chemical shift difference, normalised by the particular parameters of the exchange process. The exchange term therefore in principle contains information on thermodynamic, kinetic and structural aspects of the chemical exchange

process, and a variety of methods have been developed to extract this information, adapted to varying functional groups, molecular weights, populations and timescales (Alderson et al., 2020; Sekhar and Kay, 2019).

Although the overall $R_2$ (Eq. (1)) is readily measured by Hahn echo (HE) experiments (provided only that the timescale of exchange is much shorter than the echo duration), for the analysis of exchange it is necessary to quantify the exchange term $R_{ex}$, either by direct modulation through application of rf fields, or by determination of the exchange-free

relaxation rate, $R_{2,0}$. CPMG experiments, and related $R_{1\rho}$ measurements, are a popular example of the former approach, in which the application of a train of refocusing pulses, with frequency $\nu_{CPMG} \gtrsim k_{ex}$, reduces the magnitude of the exchange contribution. Analysis of the static field and frequency dependence of this effect can be used to quantify chemical shift differences and the populations and timescales of exchange (Millet et al., 2000). CPMG experiments have now been developed for a variety of spin systems enabling the analysis of dynamics in protein backbone and sidechains, as well as

nucleic acids (Hansen et al., 2008; Juen et al., 2016; Korzhnev et al., 2004a, 2005; Loria et al., 1999; Tugarinov et al., 2020; Yuwen et al., 2019; Yuwen and Kay, 2019).

The analysis of methyl groups is particularly interesting, as these groups are well suited to the observation of high molecular weight systems through the combination of $^{13}CH_3$ labelling against a perdeuterated background, reducing dipolar relaxation pathways, with methyl TROSY-optimised pulse sequences such as the HMQC which select for slowly relaxing

coherences within the complex energy levels of the spin system (Fig. 1) (Tugarinov et al., 2003). Given this, the multiple





quantum (MQ) CPMG experiment, based on the analysis of zero quantum (ZQ) and double quantum (DQ) coherences, provides the greatest sensitivity, although the analysis is complex as dispersions depend on both $^1$H and $^{13}$C chemical shift differences (Korzhnev et al., 2004b, 2004a). In other experiments, pulse imperfections can cause unwanted mixing between energy levels, requiring the use of compensation elements (e.g. $^1$H SQ) (Yuwen et al., 2019), while in addition back-transfer

from $C_z$ coherences is inefficient, so $^{13}$C SQ measurements can be challenging without the use of complex pulse sequences or labelling schemes (Lundström et al., 2007; Skrynnikov et al., 2001; Tugarinov et al., 2020; Weininger et al., 2012). Higher order coherences can also be exploited within fully $^{13}$CH$_3$ labelled methyl groups, such as $^1$H DQ or triple quantum (TQ) CPMG experiments, which provide exquisite sensitivity to small $^1$H chemical shift differences as the effective frequency difference, $\Delta\omega$, is magnified two or three-fold, resulting in four to nine-fold increases in $R_{ex}$ (Eq. (2)) (Gopalan et

al., 2018; Yuwen et al., 2016). However, the sensitivity of these experiments is somewhat low, particularly for large systems, due to rapid relaxation through C–H dipolar interactions during the constant time (CT) CPMG relaxation period.

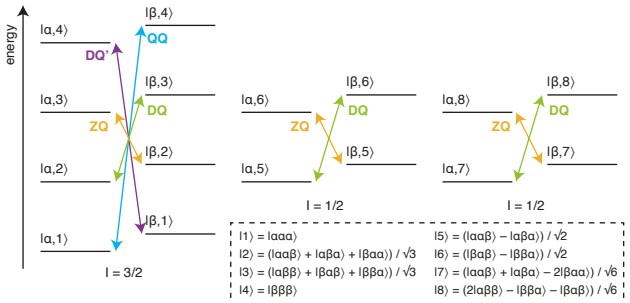

**Figure 1: Energy level diagram for an isolated $^{13}$CH$_3$ spin system. The multiple quantum transitions analysed in this work are**
**indicated with arrows. Energy levels are indexed $|C, H\rangle$ according to $^{13}$C and $^1$H spin states, where symmetrised combinations of $^1$H spins are indicated in the inset box (Tugarinov and Kay, 2006).**

An alternative approach to the detection of chemical exchange is the analysis of the relaxation and cross-correlated relaxation of multiple quantum coherences, $R_{MQ}$ and $\Delta R_{MQ}$, defined as the sum and difference of ZQ and DQ relaxation rates (Fig. 1):

$$R_{MQ} = \frac{R_{ZQ} + R_{DQ}}{2}$$


$$\Delta R_{MQ} = \frac{R_{ZQ} - R_{DQ}}{2} \tag{4}$$

Originally measured in amide spin systems, multiple quantum cross-correlated relaxation in particular was identified as being sensitive to chemical exchange through correlated fluctuations in the component proton and nitrogen frequencies, as well as being sensitive to correlated fluctuations between anisotropic chemical shifts due to rotational diffusion (Kloiber and Konrat, 2000; Tessari and Vuister, 2000). CPMG pulse trains may be combined with these measurements to identify the

contributions to $\Delta R_{MQ}$ due exclusively to chemical exchange (Dittmer and Bodenhausen, 2004). Leveraging the multi-spin nature of MQ coherences, measurements of MQ cross-correlated relaxation have since been applied to other extended spin



systems, such as adjacent Cα and Cβ spins, adjacent Cα$^{(i)}$ and Cα$^{(i-1)}$ spins, and $^{15}$N spins separated by hydrogen bonds, to detect long range conformational exchange processes (Chiarparin et al., 2001; Früh et al., 2001; Lundstrom et al., 2005).

Within methyl groups, multiple quantum relaxation and cross-correlated relaxation rates can be determined indirectly, by measurement of ZQ and DQ relaxation rates using methyl TROSY HE experiments that incorporate a filter to select only the slowly relaxing inner lines (Gill and Palmer, 2011). CPMG pulse trains may again be employed to distinguish the effects of cross-correlations in chemical shift anisotropy (CSA) from chemical exchange (Toyama et al., 2016). Alternatively, deviations from an empirical correlation established between $\Delta R_{\mathrm{MQ}}$ and measurements of cross-correlated dipole-dipole relaxation between methyl protons ($\eta_{\mathrm{HH}}$, which is proportional to $S^2_{\mathrm{axis}}\tau_c$) may be used to identify methyl groups involved in chemical exchange processes (Gill et al., 2019). However, a more detailed analysis may also be carried out, wherein the magnetic field dependence of $R_{\mathrm{MQ}}$ and $\Delta R_{\mathrm{MQ}}$ rates is analysed, and the field-dependent contributions from the $^1$H and $^{13}$C CSAs measured and subtracted to determine the contributions due to chemical exchange (Toyama et al., 2017) (re-cast from the original according to Eq. (3)):

$$R_{\mathrm{MQ,ex}} = (\xi_C^2 + \xi_H^2)B_0^2$$
$$\Delta R_{\mathrm{MQ,ex}} = 2\xi_C\xi_H B_0^2 \tag{5}$$

These expressions may be used to compare the calculated $R_{\mathrm{MQ,ex}}$ and $\Delta R_{\mathrm{MQ,ex}}$ terms with the chemical shift differences to some known reference state, assuming that all spins are part of the same exchange process and so have identical excited state populations and exchange rates(Toyama et al., 2017). However, as the expressions are symmetric in $\xi_C$ and $\xi_H$, these parameters themselves cannot be determined unambiguously. While it is possible to determine the relative sign of $\xi_C$ and $\xi_H$ from the sign of $\Delta R_{\mathrm{MQ,ex}}$, additional approaches are also required to determine their absolute signs (Auer et al., 2010; Bouvignies et al., 2010; Skrynnikov et al., 2002).

In this paper, we extend these analyses to higher order coherences accessible within methyl spin systems, and specifically to the quadruple quantum (QQ) and four-spin double quantum (DQ') transitions (Fig. 1). In this context, it is helpful to reframe the earlier analysis of the magnetic field dependence of $R_{\mathrm{MQ}}$ and $\Delta R_{\mathrm{MQ}}$ rates in terms of the ZQ and DQ rates that are directly observed in HE experiments. The magnetic field dependence of these rates all have similar functional forms (Table 1), but the DQ' and QQ rates have significantly higher sensitivity to $^1$H chemical shift differences (akin to the $^1$H TQ CPMG experiment (Yuwen et al., 2016)). We show that the four-spin relaxation rates $R_{\mathrm{DQ'}}$ and $R_{\mathrm{QQ}}$ can be measured with high sensitivity, and that analysis of the magnetic field dependence of these rates together with $R_{\mathrm{ZQ}}$ and $R_{\mathrm{DQ}}$ breaks the degeneracy of solutions and allows $\xi_C$ and $\xi_H$ to be determined unambiguously up to an overall sign. We also report improved experiments for control measurements of $^1$H and $^{13}$C CSAs, and finally demonstrate the simultaneous analysis of field-dependent HE relaxation rates with CPMG relaxation dispersion measurements.





## 2 Results

### 2.1 Theory

Transverse relaxation rates of methyl ZQ, DQ, DQ' and QQ transitions were calculated using RedKite (Bolik-Coulon et al., 2020), in the macromolecular limit assuming rapid rotation about the three-fold methyl symmetry axis and including [1]H and [13]C CSAs and interactions with external protons (Table 1). We observe that for both four-spin coherences, DQ' and QQ, relaxation due to intra-methyl dipolar interactions is zero, in common with the ZQ and DQ coherences traditionally selected in methyl TROSY experiments. This contrasts with [1]H TQ coherences used previously in CPMG experiments (Yuwen et al., 2016), which are relaxed strongly by intra-methyl dipolar interactions between [1]H and [13]C spins. However, in common with [1]H TQ coherences, the DQ' and QQ transitions remain sensitive to relaxation from dipolar interactions with external spins, and so to minimise these contributions in this work we have used only perdeuterated, selectively methyl labelled samples.

|  | Frequency | Dipolar contribution | CSA contribution | Exchange contribution |
|---|---|---|---|---|
| ZQ | $\omega_C - \omega_H$ | $\left(\frac{1}{5}b_{CX}^2 + \frac{11}{20}b_{HX}^2 - \frac{2}{5}b_{CX}b_{HX}\right)\tau_c$ | $\frac{4}{45}(\gamma_C\Delta\sigma_C - \gamma_H\Delta\sigma_H)^2 B_0^2 S_{\text{axis}}^2 \tau_c$ | $(\xi_C - \xi_H)^2 B_0^2$ |
| DQ | $\omega_C + \omega_H$ | $\left(\frac{1}{5}b_{CX}^2 + \frac{11}{20}b_{HX}^2 + \frac{2}{5}b_{CX}b_{HX}\right)\tau_c$ | $\frac{4}{45}(\gamma_C\Delta\sigma_C + \gamma_H\Delta\sigma_H)^2 B_0^2 S_{\text{axis}}^2 \tau_c$ | $(\xi_C + \xi_H)^2 B_0^2$ |
| DQ' | $\omega_C - 3\omega_H$ | $\left(\frac{1}{5}b_{CX}^2 + \frac{39}{20}b_{HX}^2 - \frac{6}{5}b_{CX}b_{HX}\right)\tau_c$ | $\frac{4}{45}(\gamma_C\Delta\sigma_C - 3\gamma_H\Delta\sigma_H)^2 B_0^2 S_{\text{axis}}^2 \tau_c$ | $(\xi_C - 3\xi_H)^2 B_0^2$ |
| QQ | $\omega_C + 3\omega_H$ | $\left(\frac{1}{5}b_{CX}^2 + \frac{39}{20}b_{HX}^2 + \frac{6}{5}b_{CX}b_{HX}\right)\tau_c$ | $\frac{4}{45}(\gamma_C\Delta\sigma_C + 3\gamma_H\Delta\sigma_H)^2 B_0^2 S_{\text{axis}}^2 \tau_c$ | $(\xi_C + 3\xi_H)^2 B_0^2$ |

**Table 1: Theoretical transverse relaxation rates of selected multiple quantum methyl transitions. Expressions for relaxation rates are divided into dipolar, CSA and exchange contributions, and transitions are labelled as indicated in Fig. 1. Dipolar contributions should be summed over external protons, where $b_{AX}$ is the dipolar coupling between a methyl spin A and an external proton X, $b_{AX} = \frac{\mu_0 \hbar \gamma_A \gamma_X}{4\pi r_{AX}^3} P_2(\cos\theta_X)$, and $P_2(x) = \frac{1}{2}(3x^2 - 1)$. Exchange contributions calculated in the fast exchange limit, where the normalised chemical shift differences $\xi$ are defined in Eq. 3.**

Contributions of chemical exchange to relaxation rates have been calculated in the limit of fast exchange (Table 1). As observed previously for the relaxation and cross-correlated relaxation of MQ coherences (Toyama et al., 2017), total relaxation rates therefore have a quadratic dependence on the static field strength, $B_0$, and this may be measured to determine the sum of the CSA and exchange terms. Therefore, control measurements of [1]H and [13]C CSA values, together with the effective rotational correlation time, $S_{\text{axis}}^2 \tau_c$, are required to isolate the contribution due only to chemical exchange.

Exchange contributions to relaxation are written in the form of squared linear combinations of normalised [1]H and [13]C chemical shift differences (Table 1). As discussed earlier, the symmetry between $\xi_C$ and $\xi_H$ in ZQ and DQ coherences precludes determination of their absolute values, even up to the overall ambiguity in sign arising from the squares. However, the four spin DQ' and QQ coherences are substantially (effectively nine-fold) more sensitive to [1]H chemical shift differences. This not only provides greater sensitivity to small chemical shift perturbations, as exploited in TQ CPMG measurements (Yuwen et al., 2016), but by breaking the symmetry between [1]H and [13]C terms it becomes possible to



determine $\xi_C$ and $\xi_H$ unambiguously (up to an overall sign) from a combined analysis of the field dependence of ZQ, DQ, DQ' and QQ relaxation rates.

**2.2 A Hahn echo super-experiment for the measurement of relaxation rates of four spin coherences**

We have developed a Hahn echo (HE) pulse sequence for the measurement of transverse relaxation rates DQ' and QQ coherences (Fig. 2A). This is adapted from the methyl TROSY-optimised $^1$H TQ CPMG experiment (Yuwen et al., 2016), with the omission of $^{13}$C 90° pulses flanking the relaxation period such that the magnetisation at the beginning of this period is $8C_y\{H_xH_yH_y\} = \frac{1}{2}i(C_+ - C_-)(3H_-H_-H_- + 3H_+H_+H_+ + \{H_+H_-H_-\} + \{H_-H_+H_+\})$, where $\{...\}$ indicates summation

over cyclic permutations. This is a mixture of coherences from which $DQ'_{\pm}$ and $QQ_{\pm}$ transitions may be isolated by phase cycling of $\psi_1$ and $\psi_2$ (Fig. 2A). As demonstrated for the simultaneous acquisition of HZQC and HDQC correlation experiments (Waudby et al., 2020), by storing individual increments of the phase program and applying receiver phase cycling post-acquisition it is possible to acquire P- and N-type pathways for both DQ' and QQ relaxation measurements within a single super-experiment (Schlagnitweit et al., 2010), without loss of sensitivity.


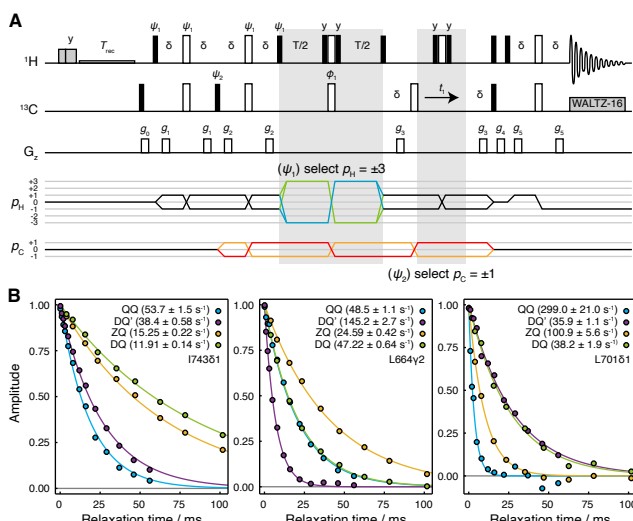

**Figure 2:** Hahn echo measurement of $R_2$ relaxation rates for methyl DQ' and QQ coherences. **(A)** Pulse sequence and coherence transfer pathways for the simultaneous measurement of DQ' and QQ Hahn echo relaxation rates. Individual steps of the phase cycles $\psi_1 = (0°, 51.4°, 102.9°, 154.3°, 205.7°, 257.1°, 308.6°)_3$, $\psi_2 = 0°_7, 120°_7, 240°_7$, are stored and processed post-acquisition to
select P- and N-type DQ' and QQ coherence transfer pathways. $T$ represents the relaxation time, and the delay δ = 1/4$J$ = 2 ms. Shaded grey pulses represent consecutive 5 kHz 2 ms $H_x$ and 3 ms $H_y$ purge pulses before the recycle delay, $T_{rec}$ (with low power presaturation applied, if required). $\phi_1$ = x, -x; $\phi_{rx}$ = x. Trapezoidal gradients were applied with powers and lengths: $g_0$ 11 G cm$^{-1}$, 1 ms; $g_1$ 14 G cm$^{-1}$, 400 μs; $g_2$ 11 G cm$^{-1}$, 200 μs; $g_4$ -14 G cm$^{-1}$, 300 μs; $g_4$ -27 G cm$^{-1}$, 500 μs; $g_5$ -44 G cm$^{-1}$, 700 μs. **(B)** HE relaxation measurements of four spin DQ' and QQ coherences for methyl groups in ILV-labelled FLN5 (800 MHz, 283 K). Measurements of
ZQ and DQ HE relaxation rates are also shown and were acquired using established methods (Gill and Palmer, 2011). Values of fitted relaxation rates are shown in the legends, with uncertainties indicating the standard error derived from fitting.



The sequence is demonstrated here using a sample of $[^2H,^{13}CH_3$-ILV]-labelled FLN5, the fifth immunoglobulin domain from the *Dictyostelium discoideum* filamin protein, comprising 108 residues. Measurements of relaxation rates were carried out at four static field strengths (600, 700, 800 and 950 MHz), and high-quality correlation spectra were obtained (Fig. S1) and

fitted to determine DQ' and QQ relaxation rates (Tables S1,2). ZQ and DQ relaxation rates were also measured using established methods (Gill and Palmer, 2011) (Tables S3,4). Measurements for representative methyl groups are plotted in Fig. 2B. Fitted relaxation rates span a wide range of values, from approximately 10 to 300 s⁻¹. We observe no fixed ordering of the various relaxation rates, and indeed in some cases four spin relaxation rates are slower than ZQ or DQ rates (e.g. for QQ relaxation in L701CD1, Fig. 2B, right hand panel).

## 175 2.3 Measurement of methyl group dynamics and $^{13}C$ chemical shift anisotropy

We next sought to determine the CSA contribution to the observed relaxation rates (Table 1), in order to isolate the exchange contribution for further analysis. Methyl $^{13}C$ CSA values can be determined by analysis of cross-correlated relaxation between the $^{13}C$ CSA and the $^1H/^{13}C$ dipolar interaction in $^{13}C$ SQ transitions, which results in differential relaxation between inner or outer lines within the $^{13}C$ quartet (Liu et al., 2003; Tugarinov et al., 2004). $^{13}C$ CSA values have previously been

measured in isolated methyl groups using constant time $^1H$-coupled HSQC experiments in which the length of the constant time delay is varied, and the relaxation of inner or outer lines fitted to exponential functions to determine the cross-correlated relaxation rate, $\eta_C$, from which the $^{13}C$ CSA, $\Delta\sigma_C$, can be derived(Tugarinov et al., 2004):

$$\eta_C = \frac{4}{45} \frac{\mu_0 \hbar \gamma_C \gamma_H}{4\pi r_{CH}^3} \gamma_C B_0 \Delta\sigma_C S_{\text{axis}}^2 \tau_c \tag{6}$$

While this measurement strategy is effective, overlap of multiplets between adjacent methyl resonances can limit the ability

to resolve the relaxation of individual transitions. A pseudo-4D variant has been developed in which the coupling is evolved in a third frequency dimension (Toyama et al., 2017), but this can require substantial measurement time. Here we suggest an alternative approach, in which an additional relaxation delay is incorporated into a non-constant time $^1H$-coupled HSQC experiment (Fig. 3A). Based on our analysis of the standard $^1H$-coupled HSQC experiment (Waudby et al., 2021), and fully incorporating the effects of relaxation and cross-correlated relaxation throughout the pulse sequence, the entire multiplet

lineshape may be expressed as a function of the $^1H$ and $^{13}C$ chemical shifts (determined from a regular HMQC spectrum), $^1H$ and $^{13}C$ relaxation rates, the scalar coupling $^1J_{CH} \approx 125$ Hz, and the parameters of interest, $S_{\text{axis}}^2\tau_c$ and $\Delta\sigma_C$. This expression may be used to fit 2D spectra obtained at multiple relaxation delays, to estimate the parameters $S_{\text{axis}}^2\tau_c$ and $\Delta\sigma_C$ simultaneously. By avoiding constant time evolution periods, this approach provides higher sensitivity, while the parametric estimation strategy allows analysis even of highly overlapped resonances, in contrast to measurements of the decay of

individual multiplet components.





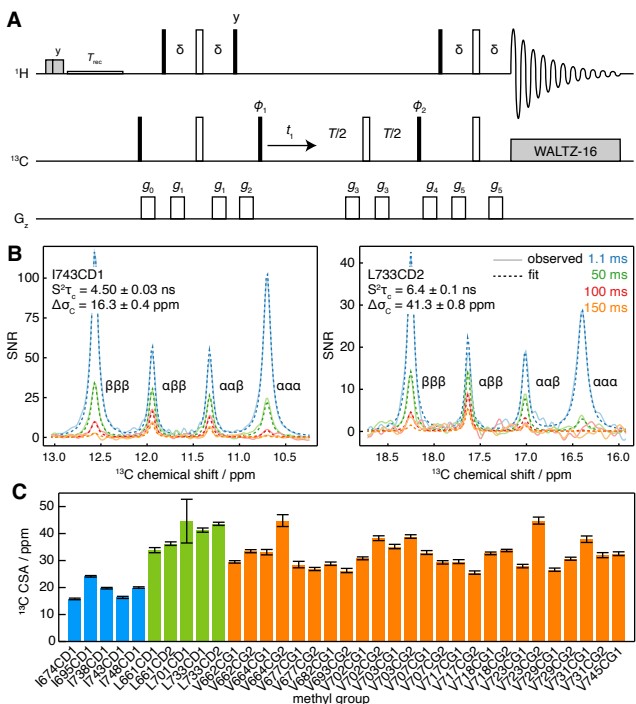

**Figure 3: Measurement of methyl $^{13}$C CSA and $S^2_{\text{axis}}\tau_c$.** (A) $^1$H-coupled HSQC with $^{13}$C Hahn echo. $T$ represents the relaxation time, and the delay $\delta = 1/4J = 2$ ms. Shaded grey pulses represent consecutive 5 kHz 2 ms $H_x$ and 3 ms $H_y$ purge pulses before the recycle delay, $T_{\text{rec}}$ (with low power presaturation applied, if required). $\phi_1 = x, -x$; $\phi_2 = x, x, -x, -x$; $\phi_{rx} = x, -x, -x, x$. Trapezoidal gradients were applied with powers and lengths: $g_0$ 25 G cm$^{-1}$, 1 ms; $g_1$ 7 G cm$^{-1}$, 1 ms; $g_2$ 9 G cm$^{-1}$, 1 ms; $g_3$ 22 G cm$^{-1}$, 300 μs; $g_4$ 18 G cm$^{-1}$, 1 ms; $g_5$ 16 G cm$^{-1}$, 1 ms. States-TPPI quadrature detection in $t_1$ was achieved by incrementation of $\phi_1$. (B) Cross-sections through $^{13}$C multiplets of isolated FLN5 methyl groups measured according to the pulse sequence in panel A, with relaxation delays as indicated, acquired at 283 K, 950 MHz ($^1$H Larmor frequency). Cross-section through the fitted spectra are also shown in dashed lines, with the fitted values of $S^2_{\text{axis}}\tau_c$ and $\Delta\sigma_C$ indicated. (C) Measured values of methyl $^{13}$C CSA in FLN5, 283 K, coloured according to the residue type. Error bars indicate the standard error determined from fitting.

Measurements were acquired for ILV-labelled FLN5 at 800 MHz, using four relaxation delays from 1 to 150 ms. Clusters of overlapping multiplets were detected and the pseudo-3D data for each cluster was then fitted to analytical expressions for the lineshape to determine $S^2_{\text{axis}}\tau_c$ and $\Delta\sigma_C$. Our observations fitted closely to theoretical expectations, and for well-resolved resonances fits can be visualised as one-dimensional cross-sections (Fig. 3B), while fits of multiple overlapping resonances are shown in Fig. S2. Values of $S^2_{\text{axis}}\tau_c$ determined in this manner were in good agreement with control measurements of the cross-correlated relaxation-induced build-up of $^1$H TQ magnetisation (Sun et al., 2011) ($R^2 = 0.97$, Fig. S3). In line with previous reports (Tugarinov et al., 2004), $^{13}$C CSA values we determined varied depending on residue type (Fig. 3C), with mean values (± s.d.) for isoleucine of 19.2 ± 3.4 ppm, leucine of 39.9 ± 4.7 ppm, and valine 32.4 ± 5.2 ppm. Full results, including $S^2_{\text{axis}}\tau_c$, are presented in Table S5.



## 2.4 Measurement of methyl $^1$H chemical shift anisotropy

The methyl $^1$H CSA, $\Delta\sigma_H$, can be determined from the differential relaxation of $^1$H doublets due to cross-correlated relaxation between the $^1$H CSA and the $^1$H/$^{13}$C dipolar interaction. This can be measured using an HMQC experiment incorporating a $^1$H Hahn echo and a filter to exclude the outer lines of the $^{13}$C multiplet (Tugarinov et al., 2004), but in

common with measurements of the $^{13}$C CSA, overlap between multiplets can be a problem. While this can again be avoided using a pseudo-4D experiment (Toyama et al., 2017), here we present a simple alternative that uses an IPAP filter to isolate $^{13}$C spin states (Fig. 4A). This allows measurement of their relaxation rates, $R_\alpha$ and $R_\beta$, from which the cross-correlated relaxation rate $\eta_H$ and the $^1$H CSA may be determined (Eq. (7)).

$$\eta_H = \frac{R_\alpha - R_\beta}{2} = \frac{4}{45} \frac{\mu_0 \hbar \gamma_C \gamma_H}{4\pi r_{CH}^3} \gamma_H B_0 \Delta\sigma_H S_{axis}^2 \tau_c \tag{7}$$


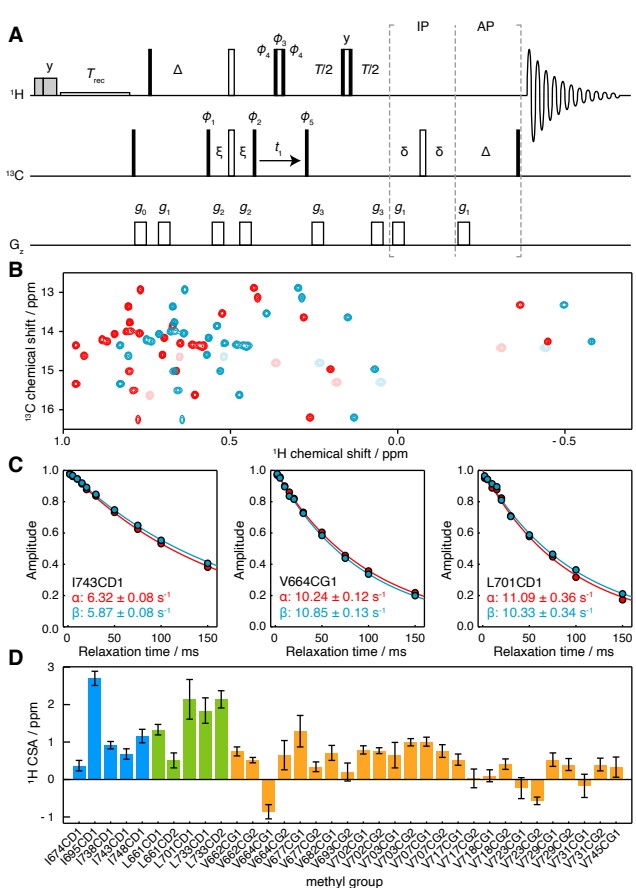

**Figure 4: Measurement of methyl $^1$H CSA values. (A) IPAP HE pulse sequence for the measurement of $^{13}$C spin state-selective $^1$H relaxation rates. Interleaved experiments are acquired using IP/AP blocks as indicated, and the sum and difference is then calculated to isolate $^{13}$C $|\alpha\rangle$ and $|\beta\rangle$ spin states. Delays $\Delta = 1/2J = 4$ ms, $\delta = 1/4J = 2$ ms, $\xi = 1/8J = 1$ ms. Shaded grey pulses**
**represent consecutive 5 kHz 2 ms $H_x$ and 3 ms $H_y$ purge pulses before the recycle delay, $T_{rec}$ (with low power presaturation applied, if required). $\phi_1 = x, -x$; $\phi_2 = (x)_4, (-x)_4$; $\phi_3 = y, y, -x, -x, -y, -y, x, x$; $\phi_4 = x, x, y, y, -x, -x, -y, -y$; $\phi_5 = x$; $\phi_{rx} = x, -x, -x, x$.**





**States-TPPI quadrature detection in $t_1$ was achieved by incrementation of $\phi_5$. Trapezoidal gradients were applied with powers and lengths: $g_0$ 31 G cm$^{-1}$, 1 ms; $g_1$ 3.9 G cm$^{-1}$, 1 ms; $g_2$ -22 G cm$^{-1}$, 300 µs; $g_3$ 16 G cm$^{-1}$, 300 µs. (B) Spectra of FLN5, 283 K, acquired at 950 MHz ($^1$H Larmor frequency) using the pulse sequence in panel A, and a relaxation delay of 2 ms. Spectra corresponding to $^{13}$C**
**$|\alpha\rangle$ and $|\beta\rangle$ spin states are plotted in red and blue respectively. Light colours indicate negative, folded signals originating from isoleucine methyl groups. (C) Spin state-selective relaxation of representative resonances, with fitted rates as indicated. (D) Measured $^1$H CSA values for FLN5, 283 K, coloured by residue type (Table S5). Error bars indicate the standard error propagated from fits to Eq. (7).**

Relaxation measurements were acquired for ILV-labelled FLN5 at 950 MHz, and high-quality sub-spectra were obtained

corresponding to $^{13}$C $|\alpha\rangle$ and $|\beta\rangle$ spin states (Fig. 4B), from which the relaxation rates $R_\alpha$ and $R_\beta$ could be measured (Fig. 4C). We note that the use of high magnetic field strengths is particularly important for these measurements given the small size of the $^1$H CSA, resulting in differences in relaxation rates typically less than 1 s$^{-1}$ (Fig. 4C). Combining these measurements with earlier measurements of $S^2_{\text{axis}}\tau_c$, $^1$H CSA values could be determined (Eq. (7)) with a precision of ca. 0.2 ppm (Fig. 4D, Table S5). $^1$H CSA values also depended on the residue type, although not as strongly as for the $^{13}$C CSA,

with mean values (± s.d.) of 0.61 ± 0.55 ppm for isoleucine, 0.79 ± 0.33 ppm for leucine, and 0.29 ± 0.28 ppm for valine.

### 2.5 Analysis of chemical exchange through field-dependent HE relaxation rates

Having carried out measurements of ZQ, DQ, DQ' and QQ HE relaxation rates in FLN5 at multiple magnetic field strengths, together with control measurements of CSA, we sought to analyse the data quantitatively in terms of chemical exchange. Assuming fast exchange, we performed linear regression of HE relaxation rates vs $B_0^2$ (Fig. 5A,B), from which measured

CSA contributions were then subtracted in order to provide estimates of $(\xi_C \pm \xi_H)^2$ and $(\xi_C \pm 3\xi_H)^2$ (Table 1). Each of these four measurements represents a pair of lines, corresponding to the positive and negative roots, in $(\xi_H, \xi_C)$ parameter space (Fig. 5C,D and S4). The points at which all four lines intersect indicate the values of $\xi_H$ and $\xi_C$ consistent with the complete set of relaxation data. The linearity of the observed relaxation rates vs $B_0^2$ (Fig. 5A,B), together with the closeness of the intersection between all four lines (Fig. 5C,D), validates our assumption of a fast exchange process underlying the

observed relaxation behaviour. We note that exchange involving a third state would also result in lines that do not intersect (discussed further in SI Text and Fig. S5).





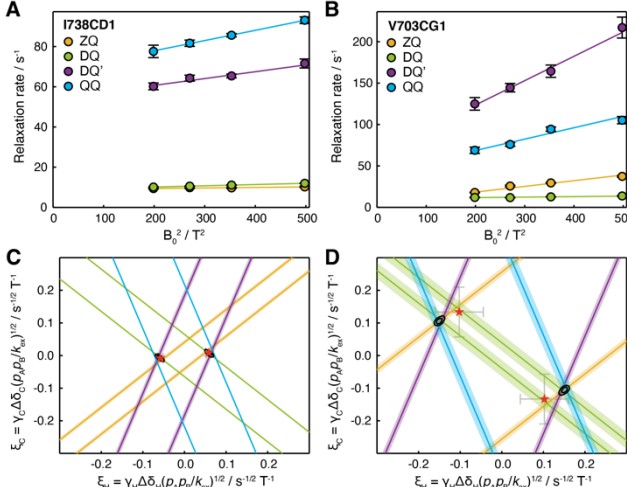

**Figure 5: Analysis of magnetic field dependence of HE relaxation rates. (A,B) Linear regression analysis of methyl ZQ, DQ, DQ'**
**and QQ HE relaxation rates as a function of $B_0^2$, for (A) I738CD1 and (B) V703CG1 resonances. Error bars indicate the standard**
**error determined from curve fitting of relaxation measurements. (C, D) Visualisation of constraints on $(\xi_H, \xi_C)$ parameter space**
**arising from HE measurements. Straight lines indicate values of $\xi_H$ and $\xi_C$ consistent with HE measurements in the upper panels,**
**calculated according to Table 1 assuming fast exchange and subtracting measured CSA contributions. Shading indicates the**
**standard error propagated from linear regression analysis and CSA measurements. Black contours indicate 68 and 95%**
**confidence intervals in $\xi_H$ and $\xi_C$, based on all four HE measurements and assuming two-state fast exchange. Red symbols indicate**
**$\xi_H$ and $\xi_C$ parameters derived from global fitting of HE and CPMG data (Fig. 6).**

To evaluate the uncertainty in the position of intersection points, $\chi^2$ surfaces can be calculated as a function of $\xi_H$ and $\xi_C$
(Fig. 5C,D, black contours). The location of the minimum on this surface for a given methyl group indicates the optimal
values of $\xi_H$ and $\xi_C$, but in the absence of additional information it is not possible to deconvolute structural information (i.e.
the chemical shift perturbations $\Delta\delta_H$ and $\Delta\delta_C$) from thermodynamic and kinetic terms. However, if several resonances are
involved in the same exchange process, then $\xi_H$ and $\xi_C$ can be compared at least in relative terms to provide structural
insight.

**2.6 Global analysis of HE relaxation and CPMG relaxation dispersion measurements**

To resolve the ambiguity in chemical shift changes, thermodynamics and kinetics inherent in $\xi$ values, we explored the joint
analysis of field-dependent HE relaxation together with MQ and [1]H SQ CPMG relaxation dispersion measurements
(Korzhnev et al., 2004b; Yuwen et al., 2019) (Fig. 6A,B). While large MQ dispersions were observed for several methyl
resonances, [1]H SQ dispersions were generally small, which indicated the potential utility of DQ' and QQ data in providing
increased sensitivity to small [1]H chemical shift differences. Inspection of individual CPMG traces indicated clearly that two
separate groups of methyl resonances, located in distinct regions of the molecule, were undergoing exchange with different
rates and populations (e.g. I743CD1 located in the G-strand, and L701CD1 located in the C/D hairpin, Fig. 6A,B).




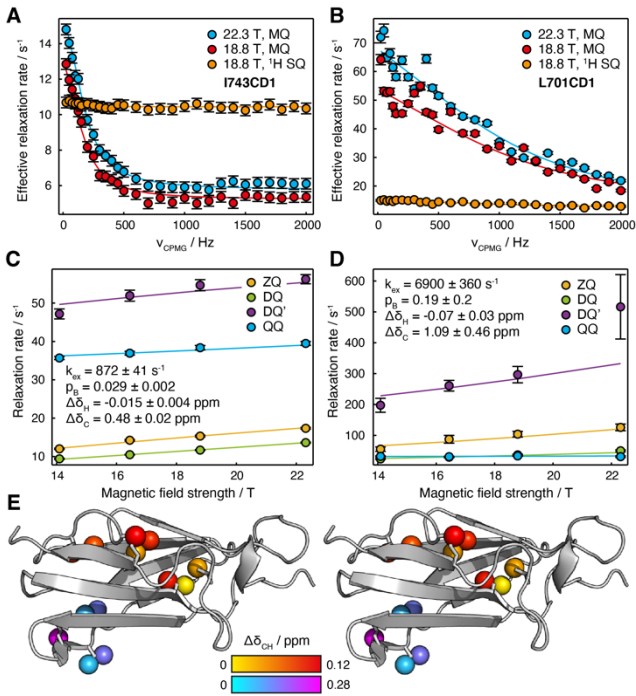

**Figure 6: Joint analysis of field-dependent HE and CPMG measurements in FLN5. (A, B)** MQ and ¹H SQ CPMG relaxation dispersion measurements (Korzhnev et al., 2004b; Yuwen et al., 2019) for **(A)** I743CD1 and **(B)** L701CD1 resonances. Error bars indicate standard errors derived from the spectrum noise. Solid lines show global fits in conjunction with HE measurements. **(C, D)** Magnetic field dependence of multiple quantum HE relaxation rates of **(A)** I743CD1 and **(B)** L701CD1 resonances. Error bars indicate standard errors determined from curve fitting. Solid lines show global fits in conjunction with CPMG measurements, with best fit parameters as shown. **(E)** Stereoview projection of fitted methyl chemical shift differences, $\Delta\delta_{CH} = \sqrt{(\Delta\delta_{CH}/4)^2 + \Delta\delta_H^2}$, on the crystal structure of FLN5 (1qfh, (McCoy et al., 1999)).

CPMG and HE data for multiple residues were fitted simultaneously to determine ¹H and ¹³C chemical shift differences, the population of the intermediate state and the exchange rate (Table S6). Relaxing our earlier assumption of fast chemical exchange (Table 1), exchange contributions to HE measurements were calculated directly from the dominant eigenvalue of the Liouvillian superoperator (Palmer and Koss, 2019). Good quality fits were obtained for all resonances, indicating the mutual consistency of HE and CPMG measurements (Fig. 6A–D and S6). HE and CPMG data for resonances within the first cluster of exchanging methyl groups, located between the A and G strands (Fig. 6E, red–yellow colouring), fitted closely to a two-state exchange model with an intermediate population of $2.9 \pm 0.2$ % and an exchange rate of $872 \pm 41$ s⁻¹ (Fig. 6A,C). Exchange within the second cluster, located around the C/D hairpin (Fig. 6E, cyan–magenta colouring), was more rapid, with a fitted exchange rate of $6900 \pm 360$ s⁻¹ (Fig. 6B,D). As expected for resonances in fast exchange (Palmer and Massi, 2006), the fitted chemical shift differences and intermediate state population were strongly correlated, which results in the large uncertainties reported. Nevertheless, for both clusters, $\xi$ values back-calculated from the fit results were consistent with our previous analysis of HE data alone (Fig. 5C,D, red symbols).



# 3 Discussion

The study of methyl groups is of fundamental importance to modern biomolecular NMR spectroscopy, both in proteins and in labelled nucleic acids (Abramov et al., 2020; Kerfah et al., 2015; Rosenzweig and Kay, 2014). Methyl $^{13}CH_3$ spin systems
provide a complex set of energy levels (Fig. 1), and in this work we have utilised the maximum range available, describing the first applications of four spin double and quadruple quantum coherences to the study of biomolecular dynamics.

      Building on earlier analyses of relaxation and cross-correlation relaxation of multiple quantum coherences (Gill et al., 2019; Gill and Palmer, 2011; Kloiber and Konrat, 2000; Toyama et al., 2017), by measuring the dependence of ZQ, DQ, DQ' and QQ HE relaxation rates on the static magnetic field strength we have demonstrated the ability to unambiguously
determine the relative chemical shift perturbations of sparsely populated intermediate states (up to an overall sign). The analysis of DQ' and QQ relaxation also provides increased sensitivity to small $^1H$ chemical shift differences. This is particularly important as these tend to be smaller than associated $^{13}C$ chemical shift differences: $^1H$ chemical shifts are determined primarily by magnetic anisotropy and ring current effects (Li and Brüschweiler, 2012), while $^{13}C$ chemical shifts are additionally and strongly affected by rotamer distributions through the γ-gauche effect (Hansen et al., 2010). We have
also demonstrated the joint analysis of HE and CPMG relaxation dispersion measurements, which provides greater detail on the thermodynamics and kinetics of the exchange process, and indeed on the existence of multiple excited states, than can be determined from analysis of HE data alone. We suggest that the combination of HE and MQ CPMG measurements may be a particularly useful pairing to provide precise estimates of both $^1H$ and $^{13}C$ chemical shift differences while avoiding insensitive DQ or TQ $^1H$ CPMG experiments (Gopalan et al., 2018; Yuwen et al., 2016). However, HE measurements alone
may also be useful in extending the analysis of dynamics in methyl groups beyond the timescales accessible to CPMG experiments.

      It is interesting to observe that the four spin DQ' and QQ transitions, in common with the inner ZQ and DQ 'methyl TROSY' transitions (Fig. 1), have zero relaxation due to intra-methyl dipolar interactions (Table 1). While the four spin transitions are certainly more sensitive to relaxation through dipolar interactions with protons from other methyl groups, this
effect nevertheless leads to an unexpectedly low exchange-free relaxation rate, particularly in comparison to $^1H$ DQ or TQ transitions previously used in CPMG experiments (Gopalan et al., 2018; Yuwen et al., 2016), which in turn leads to greater sensitivity to the effects of chemical exchange.

      We have described a new super-experiment (Schlagnitweit et al., 2010) for the simultaneous measurement of these DQ' and QQ relaxation rates (Fig. 2A). This approach, in which individual phase cycle steps are stored during acquisition
and re-combined during processing, was also recently applied to the simultaneous measurement of HZQC and HDQC correlation spectra (Waudby et al., 2020). In both instances, the total time required to acquire the pair of experiments is halved without any compromise to the quality of the individual measurements. Indeed, the IPAP scheme we have described here for measurement of the $^1H$ CSA (Fig. 4A) can also be regarded as a form of super-experiment, containing two individual experiments for the measurement of spin-state dependent $^1H$ relaxation rates. The traditional on-the-fly





implementation of receiver phase cycling during acquisition is simply a relic of the high cost of data storage in the past, with no practical benefits, and we would echo calls for the storage of individual free induction decays for processing post-acquisition to become more systematically implemented in modern software (Schlagnitweit et al., 2012).

Quadruple quantum coherences have been utilised previously in biomolecular NMR for the identification of methyl resonances in $^1$H,$^{13}$C correlation spectra, via heteronuclear quadruple quantum correlation (HQQC) and quadruple quantum-

filtered HMQC experiments (Diercks et al., 1998; Kessler et al., 1991). Their effective high gyromagnetic ratio ($3\gamma_H + \gamma_C$) has also been used to increase the sensitivity of stimulated echo diffusion measurements (Zheng et al., 2009). However, we can envisage and are currently exploring further applications of these coherences to the analysis of dynamics. Given the favourable relaxation properties of these transitions (at least, in perdeuterated molecules), HQQC experiments may prove useful in enhancing the sensitivity of titration measurements to chemical exchange, complementing our earlier work on the

two-dimensional lineshape analysis of HSQC, HMQC, HZQC and HDQC experiments (Waudby et al., 2020). Similarly, we expect that the four spin analogue of the HMQC, in which a mixture of DQ' and QQ coherences are evolved during $t_1$, may provide a useful complement to HMQC and HSQC experiments, as well as $^{13}$C-detected SQ, DQ and TQ experiments, in determining the absolute sign of the chemical shift differences to sparsely populated intermediate states (Gopalan and Vallurupalli, 2018; Skrynnikov et al., 2002).

Finally, we anticipate further developments will emerge in the analysis of field-dependent HE relaxation rates. We have demonstrated the joint analysis of HE data with CPMG measurements for residues undergoing two-state exchange (Fig. 6), but applications to more complex multi-state mechanisms should be possible using the same approach, based on analysis of the Liouvillian superoperator. Analysis in combination with other experiment types should also be possible, for example adiabatic relaxation dispersion (Chao et al., 2019). Further, while in this work we have focussed only on the gradients

obtained from regression of relaxation rates with respect to $B_0^2$, the intercepts (i.e. at zero field) also contain information on relaxation rates in the absence of exchange that could be used to constrain analyses of CPMG or similar data (Phan et al., 1996; Wang et al., 2001). Even in systems such as amides restricted to two spin coherences, ZQ and DQ HE data provide restraints that we have now demonstrated can be applied in conjunction with CPMG measurements. SQ HE measurements, which we have not explored in this work, could also provide additional restraints and remove the ambiguity of multiple

solutions illustrated in Fig. 5C,D. Together, these methods may provide new approaches for the analysis of correlated motions in extended systems, for example across nucleic acid base pairs (Chiarparin et al., 2001), within protein sidechains (Früh et al., 2001), or between adjacent residues in polypeptide backbones (Lundstrom et al., 2005).

## 4 Conclusions

Quadruple quantum and four-spin double quantum coherences in $^{13}$CH$_3$-labelled methyl groups can be easily generated from

equilibrium magnetisation, and are protected against relaxation by intra-methyl dipolar interactions, leading to unexpectedly low exchange-free transverse relaxation rates in perdeuterated macromolecules. In contrast however, these high order





coherences are highly sensitive to relaxation through chemical exchange processes, and particularly to $^1$H chemical shift perturbations. The combination of these effects means that these coherences provide near-ideal probes of conformational exchange, which we have investigated in this study using a newly developed suite of pulse sequences. Analysis of the magnetic field dependence of multiple quantum relaxation rates provides a sensitive indicator of chemical exchange, and we have shown that the combined analysis of multiple such measurements can accurately determine relative $^1$H and $^{13}$C chemical shift perturbations, up to an overall sign. We have further demonstrated that this analysis may be combined with established CPMG relaxation dispersion measurements, providing increased confidence in chemical shift perturbations together with kinetic and thermodynamic descriptions of the exchange process. Indeed, the combination of field-dependent Hahn echo relaxation measurements with relaxation dispersion measurements that we have demonstrated here for the first time may have more general applicability beyond methyl groups, improving the unique ability of NMR spectroscopy to characterise conformational exchange processes involving sparsely populated intermediate states.

## 5 Experimental

### 5.1 Sample preparation

[$^2$H,$^{13}$CH$_3$-ILV]-labelled FLN5 was expressed and purified as previously described (Cabrita et al., 2016), to yield a sample with a final concentration of 100 µM in Tico buffer (10 mM d8-HEPES, 30 mM NH$_4$Cl, 12 mM MgCl$_2$, pH 7.5, 100% D$_2$O).

### 5.2 NMR spectroscopy

NMR measurements were acquired using Bruker Avance III HD spectrometers running Topspin 3.5pl6, equipped with cryoprobes and operating at $^1$H Larmor frequencies of 500, 600, 700, 800 and 950 MHz. Data were acquired at 283 K, which was calibrated between spectrometers using a sample of d4-methanol (Findeisen et al., 2007). $^1$H,$^{13}$C correlation spectra were typically acquired with a sweep width of 16 ppm and acquisition time of ca. 100 ms in the direct dimension, and a sweep width of 15 ppm (or, if isoleucine resonances are folded, 8 ppm) and acquisition time of ca. 35 ms in the indirect dimension, centred at offsets of 0.4 and 16.7 ppm respectively. Data were processed on NMRbox (Maciejewski et al., 2017) using nmrPipe (Delaglio et al., 1995), viewed using Sparky (Lee et al., 2015), and analysed using FuDA (https://www.ucl.ac.uk/hansen-lab/fuda/) and Julia 1.5 (Bezanson et al., 2017) using the NMRTools.jl package. The assignment of FLN5 methyl resonances were obtained from the BMRB (entry 15814) (Hsu et al., 2009).

### 5.2.1 Measurement of Hahn echo relaxation rates

Hahn echo measurements of QQ and DQ' relaxation rates were acquired at 600, 700, 800 and 950 MHz using the pulse sequence described in Fig. 2A. Recycle delays of 1 s (700 MHz, 800 MHz, 950 MHz) and 1.5 s (600 MHz) were used, with 4.5 kHz WALTZ-16 $^{13}$C decoupling during acquisition. A single scan was recorded for each of the 21 steps in the $\psi_1$ and $\psi_2$ phase cycle, before looping over relaxation delays and then the $^{13}$C chemical shift evolution. Measurements were acquired





for relaxation times: 0.1, 1, 2, 3.5, 5.5, 8, 11, 15, 20, 26, 33, 41, 50 and 60 ms (600 MHz); 0.1, 1, 2, 4, 7, 11, 16, 22, 29, 37, 46 and 56 ms (700 MHz); 0.1, 1, 2, 3, 5, 8, 12, 16, 22, 29, 37, 46 and 56 ms (800 MHz); and 0.1, 1, 2, 3, 5, 7, 10, 13, 16, 22, 29, 37, 46 and 56 ms (950 MHz). Receiver phase cycling was applied following acquisition using a Julia script to select DQ'
and QQ coherence transfer pathways (Fig. 4A, Listing S2). The resulting data were processed with linear prediction and cosine-squared window functions, and peak amplitudes were then fitted to exponential functions using FuDA to determine the relaxation rates.

Hahn echo measurements of ZQ and DQ relaxation rates were acquired at 600, 700, 800 and 950 MHz, using previously described experiments, without $^{13}$C polarisation enhancement (Gill and Palmer, 2011). A recycle delay of 1 s was
used, and 20 scans were recorded at each point. Relaxation delays were set as multiples of 3.91 ms (1/2$J$, $J$ = 128 Hz): 1, 2, 4, 6, 9, 12, 16, 20, 26, 32, 40, 50, 60 × 3.91 ms (600 MHz); 1, 2, 4, 6, 9, 12, 16, 20, 26, 32, 40, 50 × 3.91 ms (700 MHz, 800 MHz); and 1, 2, 3, 4, 6, 8, 10, 12, 16, 20, 26, 32, 40 and 50 × 3.91 ms (950 MHz). Data were processed and fitted in FuDA as above to determine relaxation rates.

**5.2.2 Measurement of chemical shift anisotropies**

Measurements of methyl $^{13}$C CSA and $S^2_{\text{axis}}\tau_c$ were acquired at 800 MHz using the pulse sequence described in Fig. 3A. A recycle delay of 1.5 ms was used, and two scans were acquired, with a 56 ms acquisition time and 15 ppm sweep width for the indirect dimension, and relaxation delays of 1.1, 50, 100 and 150 ms. Data were processed with a cosine-squared window function in the direct dimension, and linear prediction and 10 Hz exponential line broadening in the indirect dimension. A list of peak positions was prepared from a $^1$H,$^{13}$C HMQC experiment, and parsed to determine clusters of non-overlapping
resonances based on a 0.05 ppm strip width in the $^1$H dimension and a 2.4 ppm strip width in the $^{13}$C dimension. For each peak within a cluster, 2D spectra of $^{13}$C quartets were simulated as recently described (Waudby et al., 2021), incorporating the additional relaxation period, $T$, shown in Fig. 2A:

$$y(\omega_H, \omega_C, T) = A \cdot \mathcal{L}(\omega_H; \omega_{0,H}, R_{2,H}) \cdot$$
$$[I_{\text{outer}} \exp(-R_{\alpha\alpha\alpha}T)\,\mathcal{L}(\omega_C; \omega_{0,C} + 3\pi J_{CH}, R_{\alpha\alpha\alpha}) +$$
$$I_{\text{inner}} \exp(-R_{\alpha\alpha\beta}T)\,\mathcal{L}(\omega_C; \omega_{0,C} + \pi J_{CH}, R_{\alpha\alpha\beta}) +$$
$$I_{\text{inner}} \exp(-R_{\alpha\beta\beta}T)\,\mathcal{L}(\omega_C; \omega_{0,C} - \pi J_{CH}, R_{\alpha\beta\beta}) +$$
$$I_{\text{outer}} \exp(-R_{\beta\beta\beta}T)\,\mathcal{L}(\omega_C; \omega_{0,C} - 3\pi J_{CH}, R_{\beta\beta\beta})]$$
(8)

where $\mathcal{L}(\omega; \omega_0, R)$ describes a Lorentzian signal with resonance frequency $\omega_0$ and relaxation rate $R$ observed at a frequency $\omega$; $I_{\text{outer}} = 3 + 3\Delta$ and $I_{\text{inner}} = 3 - \Delta$; $\Delta = e^{-4\eta_{\text{HHHH}}\tau} \cosh 2\eta_{\text{HHHC}}\tau$; $\tau = 1/(2J_{CH})$; $R_{\alpha\alpha\alpha} = R_{2,C} + 3\eta_{\text{CHCH}} + 3\eta_{\text{CHC}}$, $R_{\alpha\alpha\beta} =$
$R_{2,C} - \eta_{\text{CHCH}} + \eta_{\text{CHC}}$, $R_{\alpha\beta\beta} = R_{2,C} - \eta_{\text{CHCH}} - \eta_{\text{CHC}}$ and $R_{\beta\beta\beta} = R_{2,C} + 3\eta_{\text{CHCH}} - 3\eta_{\text{CHC}}$ ; $\eta_{\text{HHHH}} = \frac{9}{40}\left(\frac{\mu_0}{4\pi}\right)^2 \frac{\hbar^2\gamma_H^4}{r_{HH}^6} S^2_{\text{axis}}\tau_c$ ; $\eta_{\text{HHHC}} = \frac{1}{5}\left(\frac{\mu_0}{4\pi}\right)^2 \frac{\hbar^2\gamma_C\gamma_H^3}{r_{CH}^3 r_{HH}^3} S^2_{\text{axis}}\tau_c$; $\eta_{\text{CHCH}} = \frac{2}{45}\left(\frac{\mu_0}{4\pi}\right)^2 \frac{\hbar^2\gamma_C^2\gamma_H^2}{r_{CH}^6} S^2_{\text{axis}}\tau_c$; and $\eta_{\text{CHC}} = \frac{4}{45}\left(\frac{\mu_0}{4\pi}\right) \frac{\hbar\gamma_C\gamma_H}{r_{CH}^3}\gamma_C B_0 \Delta\sigma_C S^2_{\text{axis}}\tau_c$. Multiplets within each cluster were fitted as a function of the $^1$H and $^{13}$C relaxation rates, $S^2_{\text{axis}}\tau_c$, $\Delta\sigma_C$, and the scalar coupling $J_{CH}$.



$S_{\mathrm{axis}}^2 \tau_c$ were also measured at 500 MHz via the build-up of $^1$H TQ magnetisation (Sun et al., 2011), with delays of 2, 6, 12, 20, 30, 40, 50 and 60 ms used for both SQ relaxation and TQ build-up measurements. Peak amplitudes were measured

using FuDA and fitted as described (Sun et al., 2011) to determine $S_{\mathrm{axis}}^2 \tau_c$ values.

$^1$H CSA measurements were acquired at 950 MHz using the pulse sequence described in Fig. 4A. A recycle delay of 1 s was used, and 8 scans were recorded for both IP and AP experiments, before looping over relaxation delays and then $^{13}$C chemical shift evolution. Measurements were acquired for relaxation times of 2, 5, 10, 15, 20, 30, 50, 75, 100 and 150 ms. Peak amplitudes were fitted using FuDA to determine the spin state-selective relaxation rates $R_\alpha$ and $R_\beta$, from which $\Delta\sigma_{\mathrm{H}}$

was then determined according to Eq. 7.

### 5.2.3 CPMG relaxation dispersion

Multiple quantum CPMG relaxation dispersion experiments (Korzhnev et al., 2004b) were acquired at 800 and 950 MHz. A 2 s relaxation delay was used, with a 40 ms relaxation period comprising 19.2 kHz $^{13}$C pulses applied at 28 CPMG frequencies from 25 Hz to 2 kHz. Measurements were interleaved at the single scan level, alternating between high and low

power CPMG pulse trains. A $^1$H SQ CPMG measurement (Yuwen et al., 2019) was acquired at 800 MHz, with a 40 ms relaxation period comprising 10.4 kHz $^1$H pulses applied at 28 CPMG frequencies from 25 Hz to 2 kHz. For both experiments, peak amplitudes were fitted using FuDA in order to determine effective relaxation rates as a function of CPMG frequency.

### 5.3 Data analysis

Measurements of the HE relaxation rates at multiple magnetic fields were fitted by weighted linear regression as a function of $B_0^2$ to determine the combination of CSA and exchange contributions to the relaxation rate, assuming fast chemical exchange (Table 1, Fig. 5A,B):

$$R_{2,\mathrm{obs}} = R_{2,0} + (\beta_{\mathrm{CSA}} + \beta_{\mathrm{ex}})B_0^2 \tag{9}$$

The CSA contribution, $\beta_{\mathrm{CSA}}$, was then subtracted, based on the measurements above, to determine the pure exchange

contribution, $\beta_{\mathrm{ex}}$. Errors were propagated using standard methods, and used to restrain $(\xi_H, \xi_C)$ parameter space, $\xi_C + n\xi_H = \pm\sqrt{\beta_{\mathrm{ex}}}$, where $n = -3, -1, +1$ or $+3$ depending on the multiple quantum coherence being analysed (Fig. 5C,D). $\chi^2$ surfaces (Fig. 5C,D) were computed as a function of $\xi_H$ and $\xi_C$ as the sum across ZQ, DQ, DQ' and QQ coherences of the sum of the squares of residuals to fits of Eq. (9) as a function of $R_{2,0}$, using values of $\beta_{\mathrm{ex}}$ calculated from the specified $\xi_H$ and $\xi_C$ assuming fast chemical exchange (Table 1).

The joint analysis of HE and CPMG relaxation dispersion measurements was performed to determine values of $\Delta\delta_C$, $\Delta\delta_H$ for each methyl group observed, together with the parameters $k_{ex}$ and $p_B$ reflecting the kinetics and thermodynamics of the exchange process. HE relaxation rates were fitted as a function of the magnetic field strength:

$$R_{2,obs} = R_{2,0} + R_{\mathrm{csa}}(B_0) + R_{\mathrm{ex}}(B_0) \tag{10}$$



where the CSA contribution was calculated as before (Table 1), and the exchange contribution was calculated from the major eigenvalue of the Liouvillian superoperator (Palmer and Koss, 2019). In the case of two-state exchange analysed here,

$$R_{ex} = \Re\left[\frac{1}{2}\left(k_{ex} + i\Delta\omega - \sqrt{(k_{ex} + i\Delta\omega)^2 - 4ik_{ex}p_B\Delta\omega}\right)\right] \quad (11)$$

where $\Delta\omega$ is the (field-dependent) frequency difference of the multiple quantum coherence under consideration (Palmer and Koss, 2019). CPMG data were fitted to numerical simulations of the propagation of magnetisation through CPMG elements, assuming no pulse imperfections, with basis spaces $\{ZQ^-, ZQ^+, DQ^-, DQ^+\}$ and $\{H_x, H_y\}$ for MQ and $^1$H SQ CPMG experiments respectively. Residuals from all HE and CPMG datasets across multiple residues were weighted by their standard error and coupled to a Levenberg-Marquardt algorithm for fitting. Uncertainties in fitted parameters were determined from the curvature of the $\chi^2$ surface.

**Data availability**

Relaxation measurements, pulse sequences (Bruker format) and associated analysis scripts are provided in Supporting Information. The software used for global fitting of HE and CPMG data is available upon request.

**Author contributions**

Measurements and analyses were conceived and performed by CAW, and the paper was written by CAW and JC.

**Competing interests**

The authors declare that they have no conflicts of interest.

**Acknowledgements**

We acknowledge the use of the UCL Biomolecular NMR Centre and the MRC Biomedical NMR Centre, and thank associated staff for their support. This work was supported by the BBSRC (BB/T002603/1), and by the Francis Crick Institute through provision of access to the MRC Biomedical NMR Centre. The Francis Crick Institute receives its core funding from Cancer Research UK (FC001029), the UK Medical Research Council (FC001029), and the Wellcome Trust (FC001029). This study made use of NMRbox: National Center for Biomolecular NMR Data Processing and Analysis, a Biomedical Technology Research Resource (BTRR), which is supported by NIH grant P41GM111135 (NIGMS).



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
