# Peer review of "Analysis of Conformational Exchange Processes using Methyl-TROSY-Based Hahn Echo Measurements of Quadruple-Quantum Relaxation"

_Magnetic Resonance, 2021_

## Author Response (AR1)

Dear Editor,

Thank you for submitting a fine article to *Magnetic Resonance*! Both reviewers acknowledge its importance and the careful presentation and analysis of the experimental work. Its impact could be increased further by making the original NMR spectra and the data processing software used for global fitting available in a publicly accessible repository (as per the policy of *Magn. Reson*. For example, the Zenodo repository operated by CERN has proven popular.) I am looking forward to the submission of a revised version that takes all the reviewers' comments into account

Thank you and the reviewers for the careful reading of our manuscript and the thoughtful and helpful comments. We are submitted a revised manuscript that we hope fully takes these comments into account, as detailed in our interactive discussion and outlined below. Please note that line numbers refer to the track changes document.

We have now uploaded raw data and software to Zenodo repositories as suggested.

Yours, with best wishes,

Chris Waudby and John Christodoulou
* * *
**Reviewer 1**

In the manuscript, Waudby and Christodoulou present a novel NMR strategy to probe chemical exchange processes on the order of microsecond to millisecond by analyzing the static magnetic field-dependence of the relaxation rates of a set of multiple-quantum (MQ) coherence of side-chain $^1$H-$^{13}$C methyl groups. The proposed experimental strategy is based on the previously reported approaches analyzing double quantum (DQ) and zero quantum (ZQ) coherences of methyl $^1$H-$^{13}$C correlations, as developed by Toyama et al. (Nat. Commun. 2017) and Gill et al (JBNMR 2011, JBNMR 2019). In this paper, the authors extend their methods by including the magnetic field-dependence of four-spin double quantum (DQ') and quadruple quantum (QQ) coherences to break the symmetry of the $^1$H and $^{13}$C chemical shift contributions to the DQ and ZQ coherences, which enables to determine the values for the relative amplitudes of $^1$H and $^{13}$C chemical shift differences between two exchanging conformers. The authors also show that the joint analyses of magnetic-field dependence of these MQ coherences and $^{13}$C-$^1$H MQ/$^1$H SQ CPMG relaxation dispersion experiments enable the extraction of robust thermodynamic and kinetic parameters. The established methodology is successfully applied to 108-residue FLN5, the fifth immunoglobulin domain from the Dictyostelium discoideum filamin protein, and two distinct conformational exchange processes with the $k_{ex}$ of ~870 s$^{-1}$ and ~6,900 s$^{-1}$ are identified.

The paper is very well and clearly written and most of the details, including the theoretical backgrounds and pulse programs/processing scripts, are provided. The approach presented here would be extremely useful for characterizing fast exchange processes, such as microsecond-order conformational dynamics and weak ligand-binding processes, and thus would greatly accelerate investigations of various biologically and chemically important systems. The idea of observing quadruple coherence is really fascinating and would attract great interest from the broad NMR community. Therefore, I strongly recommend publication in Magnetic Resonance.

We are grateful to the reviewer for their careful reading of our manuscript, and for the helpful comments they have provided. We provide a point-by-point response to these below, to be incorporated into a revised manuscript.

I have a few suggestions for further improvement as follows.

(1) Throughout the paper, the authors assume that the conformational exchange processes affecting $^{13}$C and $^{1}$H chemical shifts are always completely correlated (i.e. they can be described with the same $k_{ex}$ and $p_B$). This should be clearly mentioned somewhere in the main text (though this is implied in lines 81-88 and line 100). I agree that this assumption is very reasonable in most cases, however, the $^{13}$C chemical shift is sensitive to the side-chain rotameric changes and $^{1}$H chemical shift is more sensitive to ring-current effects from the proximal aromatic rings, and these two processes can be attributed to two distinct conformational fluctuations with different thermodynamic and kinetic parameters. This was also suggested in the analyses of T4 lysozyme in the previous work by Toyama et al.

We assume that conformational exchange processes can be well represented by chemical exchange between discrete states, described by relative populations and forwards and backwards rate constants, and that each methyl within a state can be associated with defined 1H and 13C chemical shifts. Exchange between two states is therefore associated with both 1H and 13C chemical shift differences, both of which are perfectly correlated. However, we certainly do not exclude the possibility of additional states, associated with distinct conformational fluctuations. Indeed, we already point out (l. 311–314) that 1H and 13C chemical shifts have differential sensitivity to effects such as rotamer changes and aromatic ring currents. Where different exchange processes affect different regions of a molecule, measurements of the timescale of exchange or population of the minor state using CPMG or similar experiments can be used to identify and distinguish them (as in the present example). Where a single methyl undergoes multiple exchange processes, we also demonstrate, at least theoretically, how the analysis of field-dependent MQ HE measurements may be used to identify this (l. 253–256 and Fig. S5). However, we are happy to clarify this point in the revised manuscript, particularly in relation to the referenced work on T4 lysozyme – l. 38–40 and l. 261–262.

(2) Line 103 "While it is possible to determine the relative sign of $\delta$ C and $\delta$ H from the sign of $\Delta\delta_{MQ,ex}$, additional approaches are also required to determine their absolute signs."

Here, the authors pointed out that absolute signs of $^{13}$C and $^{1}$H chemical shift differences cannot be unambiguously determined by the previous approaches, however, the Hahn-echo analyses of ZQ, DQ', and QQ coherences presented in this paper also cannot provide the absolute signs in principle (as can be seen in two symmetric optimal values in Fig 5C and D). In that sense, this sentence may be a bit misleading.

We are happy to rephrase this in the revised manuscript (l. 106–109) – we are not trying to suggest that the present work provides an approach to determine the absolute sign.

(3) Line 126. It may be helpful to mention that the relaxation induced by the external deuterons is not considered here.

In response to a suggestion from reviewer 2, we have now revised this table to include explicit calculations of the effect of deuterons on relaxation (l. 124 and Table 1).

(4) Line 173. "We observe no fixed ordering of the various relaxation rates, and indeed in some cases four spin relaxation rates are slower than ZQ or DQ rates (e.g. for QQ relaxation in L701CD1, Fig. 2B, right hand panel)."

The QQ relaxation is the fastest in L701, I guess this should be written as L664CG2.

We are grateful to the reviewer for spotting this error – the text should refer to the four-spin double quantum transition (DQ') for L701CD1 (l. 179).

(5) Line 186. Here, the authors propose the F1-decoupled HSQC to obtain $^{13}$C CSA values. I wonder how effective this approach would be when considering the applications to more complex systems. such as high molecular proteins. In large proteins, the outer components of the quartet decay very rapidly and these outer lines are almost invisible, which might affect the accuracy of the 2D line-shape

analyses as there is less information available. Furthermore, in large systems, the signal overlap can be much more severe. In such a case, the pseudo-4D type experiment as originally proposed by Toyama et al may be a preferable.

The relative intensity of inner and outer lines depends on J(0), while the relative intensity of upfield and downfield lines depends on the product of J(0) with the 13C CSA. Because we are performing a parametric estimation (pseudo-3D lineshape analysis), the strong decay of outer lines even with no relaxation delay provides a strong constraint on J(0), which we have previously validated up to correlation times of ca. 100 ns (Waudby et al., 2021, J Magn Reson). The slowly relaxing inner lines are still readily observed for such cases, allowing access to the 13C CSA. Because peak positions can already be determined from a decoupled 2D spectrum, and 1JCH scalar couplings are more or less uniform, the issue of overlap is also not as severe a problem as might be expected: in a manuscript under preparation, we have successfully applied a similar analysis to a 45 kDa protein. However, it is true that this approach will undoubtedly reach a limit, at which point the pseudo-4D approach may be helpful to resolve overlap. Alternatively, J(0) could be measured using a separate experiment and held constant during lineshape fitting to determine the 13C CSA. We are happy to include a more detailed discussion of this in the revised manuscript (l. 350–361).

(6) Line 221 and Fig 4. Is the IP and AP scheme inverted in the pulse scheme in figure 4? Also, after the purge element, the phase of phi2 should be on y to purge the fast-relaxing outer components as these outer components evolve with J and become orthogonal with respect to the slowly-relaxing central component.

We are grateful to the reviewer for spotting these typographical errors, which have been corrected in the revised manuscript (Fig. 4 and l. 237).

(7) Line 280. Please label I743 and L701 on the structure.

We have labelled these as suggested (Fig. 6).

(8) Line 285. Regarding the best-fit parameters displayed in Figs 6 C and D, how did the authors determine the absolute sign of the 13C and 1H chemical shift differences? The absolute sign of the 13C and 1H chemical shift differences cannot be determined from the analyses of Hahn-echo relaxation measurements and 13C-1H MQ/1H-SQ CPMG dispersion experiments also do not provide the information of the sign of 13C and 1H chemical shift differences.

The absolute sign has not been determined, only the relative sign of 1H and 13C chemical shift differences. We are grateful to the reviewer for pointing this out and have added a note to the legend to clarify this important point (l. 295).

(9) Line 380. Please indicate the exact labeling pattern of "[2H, 13CH3 -ILV]-labeled" sample. I imagine the CH3/CD3 labeling for Leu/Val would be important to reduce the effects of spin flips by the external protons.

We are happy to clarify this: we have indeed used a non-stereospecific 13CH3/12CD3 labelling scheme for Leu/Val residues (l. 413).

(10) Regarding Fig S3, it would be worthwhile to mention that the slope is not exactly 1 and the offset has a non-zero value. It would be useful to comment on the potential sources of these deviations.

The fitted slope is 0.861 ± 0.096, such that the deviation of the slope from 1 is 0.139 ± 0.096 (a tension of 1.4 sigma). Similarly, the fitted offset is 0.93 ± 0.98 (a tension of 0.95 sigma). There is also a strong negative covariance between estimates of the slope and offset. Therefore, we do not believe there is evidence of significant systematic deviations between the two methods. However, we are happy to include a discussion of this point in the revised manuscript (l. 217-218).

**Reviewer 2**

This paper describes an important development of a novel Hahn echo experiment for methyl quadruple quantum coherences. The theory shows that the Hahn echo relaxation rate constant is very sensitive to exchange and breaks the problem of symmetry that prevents absolute sign determination of chemical shift differences in other Hahn echo experiments. Pulse sequence for the experiment is well-described and validated (and shows excellent sensitivity). The experimental work is very thorough and includes careful assessment of the contributions of chemical shift anisotropy to the field-dependent relaxation rates. Finally, the paper shows how joint analysis of the Hahn echo and CPMG measurements leads to more complete characterization of the exchange process. I expect this new experiment to be widely adopted in the set of powerful NMR experiments using the unique properties of methyl groups to characterize dynamics processes.

We are grateful to the reviewer for their careful reading of our manuscript and appreciate their positive and helpful feedback. We provide a point-by-point response to these points below.

It may be that the authors are already at work on extensions to this work, but it seems feasible to modify the sequence for dispersion type measurements, using for example the heteronuclear double resonance relaxation approaches of Bodenhausen and coworkers, because dephasing of longitudinal 1H operators is not an issue (as in DQ/ZQ TROSY approaches).

Yes, this is an excellent idea, and we will be reporting on such applications in a forthcoming manuscript.

Minor:

Should table 1 include contributions for relaxation from dipole interactions with remote deuterons? The dipole interaction is weaker, but in a deuterated background, there are many deuterons.

The primary aim of calculating dipolar interactions with protons was to demonstrate the absence of intra-methyl relaxation pathways, but we agree that contributions from external deuterons will be an important contribution to the total relaxation rate. For completeness, as the reviewer suggests, we have now included a calculation of these contributions in the revised manuscript (Table 1).

In the joint analysis (Fig. 6 for example), were individual CSA values used for each spin or average values for residue type? Does it matter within the experimental uncertainties shown? Do other users need to remeasure the CSA's for each protein or are average values enough?

This is a great question, which has prompted us to make estimates of the relative contribution of CSA terms, and variability in CSA, to observe relaxation rates. We find that the effect of variations in CSA between methyls is only of the order of a single percent. This suggests that it will be sufficient for most users to use average values (and indeed the variation in CSA can be incorporated into calculated uncertainties), which will help to accelerate applications of this analysis. We're grateful to the reviewer for prompting this analysis, which we have incorporated into the revised manuscript (l. 363-369).

In Fig. 6C and 6D, should the x-axis be magnetic field-squared?

No – given the additional information provided by the CPMG data, we have fitted the data using a more complete model of chemical exchange (Eq. 11) which interpolates between fast and slow exchange limits. We have added notes to the main text and figure legend to clarify this point (l. 294 and l. 298–301).

The authors have done an excellent job of referencing a now extensive literature on chemical exchange in biological molecules. A couple of other references might be of interest:

Multiple quantum relaxation outside the fast-exchange limit:

C. Wang and A. G. Palmer, Differential multiple quantum relaxation caused by chemical exchange outside the fast exchange limit, J. Biomol. NMR 24, 263-268 (2002)

Joint Hahn echo/CPMG analysis of 13C relaxation:

N. E. O'Connell, et al., Partially folded equilibrium intermediate of the villin headpiece HP67 defined by 13C relaxation dispersion, J. Biomol. NMR 45, 85-98 (2009).

We thank the reviewer for highlighting these interesting papers – we're very pleased to reference and discuss these in the revised manuscript (l. 302–304, l. 327, l. 384, l. 389).